# Carcass Characteristics, Physicochemical Properties, and Texture and Microstructure of the Meat and Internal Organs of Carrier and King Pigeons

**DOI:** 10.3390/ani10081315

**Published:** 2020-07-30

**Authors:** Dariusz Kokoszyński, Kamil Stęczny, Joanna Żochowska-Kujawska, Małgorzata Sobczak, Marek Kotowicz, Mohamed Saleh, Martin Fik, Henrieta Arpášová, Cyril Hrnčár, Karol Włodarczyk

**Affiliations:** 1Department of Animal Sciences, Faculty of Animal Breeding and Biology, UTP University of Science and Technology, 85084 Bydgoszcz, Poland; kamil.steczny@o2.pl (K.S.); karwlo19@gmail.com (K.W.); 2Department of Meat Science, Faculty of Food Sciences and Fisheries, West Pomeranian University of Technology, 71550 Szczecin, Poland; joanna.zochowska-kujawska@zut.edu.pl (J.Ż.-K.); malgorzata.sobczak@zut.edu.pl (M.S.); marek.kotowicz@zut.edu.pl (M.K.); 3Department of Poultry and Animal Production, Faculty of Agriculture, Sohag University, 82524 Sohag, Egypt; bydg2016@gmail.com; 4Department of Small Animal Science, Faculty of Agrobiology and Food Resources, Slovak University of Agriculture, 94976 Nitra, Slovakia; martin.fik@uniag.sk (M.F.); henrieta.arpasova@uniag.sk (H.A.); Cyril.Hrncar@uniag.sk (C.H.)

**Keywords:** pigeons, carcass measurements, carcass composition, meat color, electrical conductivity, texture, microstructure, digestive system

## Abstract

**Simple Summary:**

The tradition of pigeon meat consumption dates back to ancient civilizations. Today, pigeons are a popular meat in the cuisines of China, North America, North Africa, and some European countries. The aim of this study was to compare carrier pigeons and pigeons of the King breed after three reproductive seasons for carcass weight and measurements, carcass composition, physicochemical characteristics, the texture, rheological properties and microstructure of meat, and some biometric characteristics of the digestive system. Significant differences (*p* < 0.05) were observed between the pigeon groups in terms of the carcass weight and measurements, carcass composition (except for breast muscle percentage), physicochemical (except for pH_24_ and redness of breast muscles) textural (except for cohesiveness and shear force), rheological, and microstructural characteristics, and more digestive system characteristics. These differences result primarily from the type of use. King pigeons are raised for meat, and carrier pigeons are used for flying.

**Abstract:**

Pigeons have been the subject of research in the past, but the knowledge gained is incomplete and must be extended. The aim of the study was to provide information about differences in carcass weight and measurements, carcass composition, proximate chemical composition, acidity, electrical conductivity, color attributes, the texture, rheological properties and microstructure of the meat, and some biometric characteristics of the digestive system in carrier and King pigeons, and also to determine if the two compared breeds meet the expectations of pigeon meat consumers to the same extent. The study involved 40 carcasses from carrier pigeons and King pigeons after three reproductive seasons. The chemical composition was determined by near-infrared transmission (NIT) spectroscopy, color coordinates according to CIELab, the texture according to Texture Profile Analysis (TPA) and Warner–Bratzler (WB) tests, and the rheological properties of meat according to the relaxation test. The compared pigeon groups differed significantly (*p* < 0.05) in carcass weight and measurements, carcass composition (except breast muscle percentage), chemical composition (except leg muscle collagen content) and electrical conductivity, lightness (L*), yellowness (b*), chroma (C*) and hue angle (h*), textural characteristics (except cohesiveness and Warner‒Bratzler shear force), rheological properties, microstructure of the pectoralis major muscle, as well as the total length of intestine and its segments, duodenal diameter, weight of proventriculus, gizzard, liver, heart, and spleen. The sex of the birds had a significant (*p <* 0.05) effect on the carcass weight, chest circumference, carcass neck percentage, breast muscle collagen content, and caeca length. The genotype by sex interaction was significant (*p* < 0.05) for fat content, collagen content, hardness, sum of elastic moduli and sum of viscous moduli of the pectoralis major muscle, protein and collagen content of leg muscles, duodenal and caecal length, jejunal and ileal diameter, and spleen weight. The obtained results show a significant effect of genetic origin and sex on the nutritive and technological value of the meat, and on the digestive system development of the pigeons.

## 1. Introduction

Pigeons are probably the first bird species to have been reared by humans [1]. In the past, pigeons were kept for emotional, religious (as sacrificial birds) and cultural reasons, but most often they were used to carry messages [2,3]. The tradition of eating young pigeons (squabs) goes back to ancient Egypt, and in subsequent centuries pigeon breeding became widespread in Europe, North America, and Asia. Currently, the largest producer of pigeon meat in the world is China, with an annual production of around 680 million squabs, around 80% of global production [4]. Other major producers of pigeon meat include the USA and Canada. American breeders produce around 2.5 million squabs a year [5]. The top European producers of pigeon meat are Great Britain, France, and Italy [6]. Meat pigeon farming is also popular in Denmark, Germany, Spain, and Hungary [5,7,8]. Being a niche species of poultry in many countries today, pigeons are kept in flocks of several dozen birds for personal use, sometimes as a source of extra income. In many countries, keeping ornamental and carrier pigeons, which take part in organized competitions, is becoming increasingly popular as a hobby.

Worldwide, pigeon meat is obtained principally by slaughtering young birds. Squabs are ready for slaughter at 28–30 d of age and 400–700 g of body weight, depending on the breed and rearing method [3,9,10,11]. Spent pigeon meat is considered a slaughter by-product of birds that have outlived their reproductive lives. In many countries, pigeon meat is also harvested by hunting. For example, in Spain, over two million feral pigeons were annually harvested by hunting for human consumption [7].

Around 50 breeds/genotypes of meat-type pigeons are currently known, but only a few of them have a large share in the production of pigeon meat around the world. In the USA, meat pigeon production is based on the poor-flying King and Hubble breeds [12]. In Poland, the most popular breeds of meat pigeons are King, Strasser, Texan, Cauchois, Mondain, Lahore, Giant Homer, Polish Lynx, and Wrocławski Meat [3].

Despite the birds’ many advantages, pigeon carcasses, meat, and giblets are not very popular among consumers in most countries, including Poland. This is due to limited availability resulting from the small number of large-scale pigeon farms, a scarcity of pigeon slaughterhouses, a nonexistent or vanishing tradition of pigeon meat consumption, a lack of knowledge about the advantages of pigeon meat, and the relatively high price of pigeons when compared to the carcasses and meat of broiler chickens [12].

In comparison to broiler chicken carcasses, pigeon carcasses have a higher percentage of breast muscle and a much lower percentage of leg muscle [3,5]. The meat of pigeons, especially breast meat, is very nutritious because it is rich in high-value protein and low in cholesterol. However, squab meat, in particular leg meat, is characterized by a relatively high fat content, as well as high energy value [13]. The relatively high fat content of pigeon meat improves its taste, juiciness, and aroma intensity [12]. Pigeon breast and leg muscles show a favorable fatty acid profile, with a prevalence of unsaturated fatty acids and a high content of polyunsaturated fatty acids, including linoleic and linolenic acids [6,14]. Pigeon meat is a good source of vitamins and minerals, especially vitamins of the B group, phosphorus, iron, and zinc. One hundred grams of raw squab meat contains 41% of the daily human requirement for vitamin B_6_, 25% for thiamine (vitamin B_1_), 24% for riboflavin (vitamin B_2_), 44% for phosphorus, 35% for iron, and 28% for zinc [15]. Squab meat is delicate, tender, and moist [10].

The existing body of research suggests a significant effect of genotype on the basic chemical composition (content of water, protein, fat, and ash), cholesterol content of breast muscle, and lipid fatty acid profile of breast and leg muscles [6]. Another factor contributing to the chemical composition of the meat is the composition of the pigeons’ diet [16]. In turn, Bu et al. [17] pointed to the effect of age on the protein and fat content of breast muscle in King pigeons.

The lack of studies comparing carrier pigeons (used mainly for flying) and King pigeons (raised for meat production) for carcass, meat quality, and digestive system traits encouraged us to conduct this research. The aim of the study was to compare carrier pigeons and King pigeons for carcass weight and measurements, carcass tissue composition, chemical composition (water, protein, fat and collagen content) of breast and leg muscles, and physicochemical characteristics (pH_24_ and EC_24_—electrical conductivity, L*—lightness, a*—redness, b*—yellowness, C*—chroma, h*—hue angle), texture (hardness, springiness, cohesiveness, chewiness, gumminess, Warner‒Bratzler shear force), rheological properties (sum of elastic moduli, sum of viscous moduli), and microstructure (fiber cross-section area, fiber perimeter, horizontal and vertical fiber diameter, perimysium and endomysium thickness) of the pectoralis major muscle. The results obtained allowed for determining and comparing the suitability for consumers of the carcasses and meat from the examined pigeon breeds.

## 2. Materials and Methods

### 2.1. Carcass Collection and Evaluation of Carcass Traits

The materials for the study consisted of 40 carcasses collected from carrier pigeons (a variety of rock pigeon, *Columba livia*) and King pigeons after three reproductive seasons. The ratio of male to female carcasses was 1:1. Carcasses were purchased from a small breeder of pigeons. According to the information obtained from the breeder, the carcasses originated from birds culled as part of a flock management program; they were reared in a pigeon loft and fed ad libitum wholegrain or seed diet during the reproductive period. The dietary mixture contained yellow maize (26%), peas (23%), wheat (11%), red sorghum (11%), white sorghum (11%), yellow millet (5%), rapeseed (4%), black sunflower (3%), buckwheat (2%), shelled barley (2%), and shelled oats (2%). Pigeons had access to a mineral mixture (offered in a separate trough) and 24 h ad libitum access to water.

Eviscerated carcasses with necks were chilled for 18 h at 4 °C. Upon removal from the refrigerator, the carcasses were individually weighed to within 0.01 g on an electronic scale (PS 1000/X, Radwag, Radom, Poland). With a dressmaker’s tape, accurate to 1 mm, measurements were made of: carcass length (the distance from the beginning of the neck to the posterior superior tuberosity of the ischium), trunk length (the distance from the tuberosity of the shoulder joint to the posterior tuberosity of the ischium), keel length (the distance from the anterior to the posterior edge of the keel), chest circumference (behind the wings through the anterior edge of the keel and middle thoracic vertebra), and drumstick length (the distance from the knee joint to the hock joint).

Carcasses were dissected using a simplified method developed and reported by Ziołecki and Doruchowski [18]. During the dissection, each carcass was separated into the following parts: neck without skin, both wings with skin, breast muscles (pectoralis major muscle and pectoralis minor muscle), leg muscles (all the muscles from both thighs and drumstick muscles), abdominal fat, and carcass remainders, i.e., the skeleton with a certain number of muscles (intercostal, dorsal, superscapular, and others) together with the kidneys, but without other internal organs. The dissected carcass components were weighed to within 0.01 g on an electronic scale, and their percentage in the eviscerated carcass with neck was calculated.

After weighing the carcass parts, 40 breast and 40 leg muscles were sampled to determine the basic chemical composition, and 40 pectoralis major muscle samples were collected to determine color attributes, texture, rheological properties, and the microstructure.

### 2.2. Basic Chemical Composition

The water, protein, fat, and collagen contents of breast muscles (both muscles on one side of the carcass) and of the muscles of both legs were determined using a FoodScan near-infrared spectrophotometer (FoodScan Laboratory, Foss, Warrington, UK).

### 2.3. Physicochemical Characteristics

After making carcass measurements (but before dissection), the acidity (pH_24_) and electrical conductivity (EC_24_) of the pectoralis major muscle were determined. The pH of breast meat was measured with a pH-Star CPU device (Ingenieurbüro R. Matthäus, Nobitz, Germany) tipped with a glass electrode for meat pH determinations. Before pH_24_ measurement, the pH meter was calibrated using calibration buffers (pH 7.0 and pH 5.5), and later adjusted to the meat temperature of 4 °C. pH values were read from a liquid crystal display to the nearest 0.01. The electrical conductivity (mS/cm) of the pectoralis major muscle was measured with an LF-Star CPU device (Ingenieurbüro R. Matthäus, Nobitz, Germany). The electrodes of the conductivity probe were inserted into the pectoralis major muscle at an angle of 90° along the muscle fibers. The measurement was accurate to 0.1 mS/cm.

Color coordinates (L*—lightness, a*—relative redness, on red‒green axis, b*—relative yellowness, on yellow‒blue axis [19]) were determined on the inner surface of the raw pectoralis major muscle immediately after they were dissected and weighed. The L*, a*, and b* color coordinates were measured with a CR-410 chroma meter (Konica Minolta, Osaka, Japan). Wide-area illumination (measurement area 20 mm, illuminant D_65_) was used for the determination of the color coordinates. Before the measurements, the chroma meter was calibrated against a white reference tile (Y = 94.40, x = 0.3159, y = 0.3325). The redness (a*) and yellowness (b*) values were used to calculate chroma (C*) and hue angle (h*). Chroma (C*) was calculated with the formula C = (a*^2^ + b*^2^)^1/2^, and hue angle (h*) with the formula (b*/a*) tan^−1^ [20].

The pectoralis major muscle was also evaluated for textural traits, rheological properties, and microstructure.

### 2.4. Meat Texture

The textural traits (hardness, cohesiveness, chewiness, springiness gumminess, Warner‒Bratzler shear force) of the 40 samples of processed pectoralis major muscle of pigeons after three reproductive seasons were determined using an Instron 1140 machine (Instron Corp., Norwood, MA, USA). The tests were performed on heat-treated samples. The samples of meat were tightly packed into plastic bags and heated in water at 72 ± 2 °C until the internal temperature reached 70.2 °C in the geometric center of the sample. Next, the samples were chilled in water to 20 °C, wrapped in food-grade plastic film, stored for 12 h in a refrigerator at 4 °C, and heated to 18 °C. After removing the wrap from each sample of the pectoralis major muscle, slices of 20 ± 1 mm thickness were cut perpendicular to the muscle fiber orientation using a Siemens MS600 electric slicer (Hausgeräte GmbH, Köln, Germany). The prepared meat samples were measured for textural traits using the Texture Profile Analysis (TPA) and WB (Warner‒Bratzler) tests.

The TPA test involved twice pushing a plunger of 0.96 cm diameter into a sample 20 ± 1 mm high, to 80% (16 mm) of its depth. Using the force‒deformation curve relationship, the following texture parameters were calculated: hardness—maximum height of the first peak, cohesiveness—ratio of the second peak area to the first peak area, springiness—width of the base of the ascending portion of the second peak, gumminess—product of hardness and cohesiveness, and chewiness—a product of hardness, cohesiveness, and springiness [21]. For each sample, the TPA test was repeated four times and the total number of determinations was 160.

The Warner‒Bratzler (WB) test consisted of cutting, transversely across the muscle fibers, a sample with a 10 × 10 mm cross-sectional area using a triangular blade. A crosshead speed of 50 mm/min was applied. This test was used to determine the trait WB shear force.

### 2.5. Rheological Properties

The rheological properties (sum of viscous moduli and sum of elastic moduli) of the 40 samples of processed pectoralis major muscle of pigeons were determined using an Instron 1140 machine (Instron Corp., Norwood, MA, USA). During the relaxation test, a plunger of 1.26 cm diameter was pushed into the muscle samples 2 mm deep while recording for 90 s the changes in tension. The springiness and viscosity moduli were calculated using the generalized Maxwell model, consisting of three elements connected in parallel: Hooke’s body and Maxwell’s two viscoelastic bodies. The model’s equation is:(1)$$\delta =\varepsilon *\left[E_{0}+E_{1}*\exp\left(\frac{-E_{1}*t}{\mu_{1}}\right)+E_{2}*\exp\left(\frac{-E_{2}*t}{\mu_{2}}\right)\right]$$
where:*δ*—tension (kPa);*ε*—deformation;*E*_0_—elastic modulus of Hooke’s body (kPa);*E*_1_, *E*_2_—elastic moduli of Hooke’s body 1 and 2 (kPa);*µ*_1_, *µ*_2_—viscous moduli of Maxwell’s body 1 and 2 (kPa × s);*t*—time.

To make the interpretation of the results easier, the sum of elastic moduli (*E*_0_ + *E*_1_ + *E*_2_) and the sum of viscous moduli (*µ*_1_ + *µ*_2_) were calculated for each sample.

### 2.6. Microstructure of Meat

For histological analysis, samples of pectoralis major muscle were collected from 40 birds. The samples collected from the middle part of the pectoralis major muscle were fixed with Sannomiya solution, dehydrated in alcohol and benzene, and embedded in paraffin blocks. The blocks were sectioned with microtome, and sections of 10 µm were placed on glass slides. The preparations were counterstained with hematoxylin and eosin [22] and embedded in Canada balm. The microstructure of the pectoralis major muscle (fiber cross-section area, fiber perimeter, horizontal fiber diameter (H), vertical fiber diameter (V), perimysium and endomysium thickness) was determined with a MultiScanBase v. 13 image analysis system (Computer Scanning System Ltd., Warsaw, Poland). Two preparations were made and determined for each sample. Around 200 muscle fibers were measured in each preparation, and around 100 measurements of perimysium (connective tissue surrounding a muscle fiber bundle) and endomysium (connective tissue surrounding a single muscle fiber) thickness were made. A magnification of 100× was applied. Based on the data for horizontal (H) and vertical (V) diameters of the pectoralis superficialis muscle fiber, the H:V diameter ratio was calculated.

### 2.7. Digestive System Characteristics

In our experiment, we also measured the dimensions of the digestive system. A dressmaker’s tape was used to measure different intestine segments (duodenum, jejunum, and ileum collectively; caecum, colon) to the nearest 0.1 cm. The length of the duodenum was measured from the pylorus to the pancreatic loop, and the length of jejunum and ileum from the pancreatic loop to the ileocaecal junction. The length of the caeca was measured from the mouth of the ileum to the vertex of the right and left caecum. The length of the colon was measured as the distance from the mouth of the caeca to the cloaca. The diameters of individual segments—the anterior, middle, and posterior parts of duodenum, jejunum, and ileum collectively, caecum, colon—were measured with electronic calipers to the nearest 0.01 mm. During evisceration, the gizzard, proventriculus, heart, spleen, and liver were separated and weighed to within 0.01 g on an electronic scale.

### 2.8. Statistical Analysis

The numerical data obtained for the carcass weight and measurements, percentage of carcass components, basic chemical composition of breast and leg muscles, as well as physicochemical traits, texture and rheological properties, microstructure of the pectoralis major muscle, and dimensions of the digestive system of carrier and King pigeons were subjected to statistical analysis. In the first stage of the analysis, we used the Shapiro–Wilk test to determine if the empirical distribution of the traits was the same as the normal distribution. Arithmetic means as well as the standard error of the mean (SEM; collectively for both genotypes) were calculated for each tested trait. Two-way analysis of variance was used to determine the effect of genotype and sex on the analyzed carcass traits, meat quality traits, and dimensions of the digestive system. Finally, the following linear model was used: *Y_ijk_* = *μ* + *a_i_* + *b_j_* + (*a…b*)*_ij_* + *e_ijk_* where *Y_ijk_*—value of the analyzed trait, *μ*—overall mean for the tested trait, *a_i_* —effect of i-th genotype, *b*_j_—effect of j-th sex, (*a…b*)*_ij_*—genotype by sex interaction, *e_ijk_*—random error.

Statistical calculations were made using the SAS package (SAS Institute Inc., Gary, NC, USA), version 9.4 [23]. Significant differences between the arithmetic mean values of the analyzed traits of the pigeons from the compared genotypes as well as between males and females were determined using Tukey’s test. Differences were considered significant at *p* < 0.05.

## 3. Results

### 3.1. Carcass Characteristics

Carrier and King pigeons after three reproductive seasons differed in terms of the carcasses’ weights and measurements. Male and female pigeons of the King breed had significantly (*p* < 0.05) higher carcass weight, chest circumference, and lengths of carcass, trunk, keel, and drumstick compared to carrier pigeons. Regardless of genotype, males were significantly (*p* < 0.05) heavier and had a significantly greater chest circumference. The genotype by sex interaction for body weight and measurements was not significant (Table 1).

The carcasses obtained from King pigeons after three reproductive seasons showed a significantly (*p* < 0.05) higher content of leg muscle, abdominal fat, neck, wings, and carcass remainders, as well as a significantly lower percentage of skin with subcutaneous fat compared to carrier pigeons. Pigeon genotype had no significant (*p* > 0.05) effect on breast muscle percentage in eviscerated carcasses with necks. On average, male and female carrier pigeons were characterized by a higher breast muscle percentage compared to King pigeons. Male carcasses had a significantly higher neck percentage. The genotype by sex interactions for the tested slaughter traits of the pigeons was not significant (Table 2).

### 3.2. Meat Quality

Differences in the genetic origin of the pigeons had a significant (*p* < 0.05) effect on the water, protein, and fat content of breast and leg muscles and on the collagen content of breast muscle. The breast muscle of King pigeons contained significantly (*p* < 0.05) less water and significantly more (*p* < 0.05) protein, fat, and collagen. In turn, the leg muscles of King pigeons had significantly lower water and protein contents, and a significantly higher fat content compared to the leg muscles of carrier pigeons. The genotype by sex interactions was significant for the collagen content of breast and leg muscles, for the protein content of leg muscle, and for the fat content of breast muscle (Table 3).

Carrier pigeons and King pigeons also showed differences in terms of the electrical conductivity, lightness, and yellowness, chroma, and hue angle of the pectoralis major muscle. The breast muscles of King pigeons showed a significantly higher electrical conductivity, lightness. yellowness, chroma, and hue compared to those of carrier pigeons. The sex of birds had no significant effect on the physicochemical properties of the pectoralis major muscle in pigeons except for the hue angle. The genotype by sex interaction was not significant for the determined physicochemical traits (Table 4).

Pigeon genotype had a significant (*p* < 0.05) influence on hardness, springiness, chewiness, gumminess, and on the sum of elastic moduli and the sum of viscous moduli. After heat treatment, the pectoralis major muscle of King pigeons exhibited significantly (*p* < 0.05) greater hardness, springiness, chewiness, gumminess, and on the sum of elastic moduli and the sum of viscous moduli. The tenderness of the pectoralis major muscle, as determined by the maximum Warner‒Bratzler shear force, was similar (a nonsignificant difference) in carrier and King pigeons. The sex of the birds did not significantly affect the textural traits and rheological properties of the pectoralis major muscle in carrier and King pigeons. The genotype by sex interactions for hardness, the sum of elastic moduli, and the sum of viscous moduli was statistically significant (*p* < 0.05; Table 5).

The present study revealed a significant effect of pigeon genotype on the microstructural characteristics of the pectoralis major muscle. Carrier pigeons demonstrated a significantly greater fiber cross-section area, fiber perimeter, horizontal and vertical Feret diameters, and perimysium and endomysium thickness. The sex of birds and the genotype by sex interaction was not significant (Table 6).

### 3.3. Digestive System Characteristics

Carrier pigeons and King pigeons differed (*p* < 0.05) in total intestinal length and length of intestinal segments. Significant differences were also noted for the duodenal diameter and the weight of the stomach (proventriculus, gizzard), liver, heart, and spleen. King pigeons had a significantly longer total intestine, total jejunum and ileum, caeca, and colon, as well as significantly heavier proventriculus, gizzard, liver, heart, and spleen when compared to carrier pigeons. In turn, carrier pigeons exhibited greater duodenal length and diameter in comparison to King pigeons. Bird sex had a significant effect on the length of caeca. The genotype by sex interactions was significant (*p* < 0.05) for duodenal length, length of both caeca, jejunal and ileal diameter, and spleen weight (Table 7).

## 4. Discussion

The carrier and King pigeons evaluated in our study differed in carcass weight and measurements. The large differences between the compared pigeon groups are due to selective breeding of King squabs for rapid growth rate and high body weight. Kadhim and Mohamed [24] found a greater body length (male 23.2 cm, female 21.9 cm), and smaller keel length (male 8.2 cm, female 6.3 cm) in adult local homing pigeons compared to those in our study. In turn, Pawlina and Borys [25] observed greater trunk length in 24-week-old Wrocławski Meat pigeons, weighing an average of 625 g (male) and 585 g (female) compared to the carrier and King pigeons evaluated after three breeding seasons. Parvez et al. [26], who investigated 15 different breeds of pigeons obtained from 30 selected pigeon farms, found that the compared breeds differed significantly in mature body weight, body length, length of shank, head, middle toe and bill, and wingspan. Strasser pigeons were characterized by the greatest mature body weight (748.2 g), body and head lengths (41.6 cm and 8.0 cm, respectively), Homer pigeons by the greatest shank and bill lengths (3.4 cm and 2.8 cm, respectively), Jacobin pigeons by the greatest wingspan (81.0 cm), and the Pouter breed by the longest middle toe (4.4 cm). The compared pigeon genotypes were characterized by a high but similar content of breast muscle in eviscerated carcass with neck. The similar carcass breast muscle percentage in both pigeon groups tested was associated with selective breeding for high breast muscle content in King pigeons, while in carrier pigeons it was associated with the high physical activity of breast muscle related to flying. Carrier pigeons showed a significantly higher content of skin with subcutaneous fat, which is physiologically determined. A thick layer of fat and feathers protect pigeons from low temperatures, especially during flight. Fat provides a reserve of energy and water (formed during fat burning), which are used during flight [27]. Jiang et al. [4] observed lower breast muscle content in 28-day-old White King pigeons compared to the studied genotypes, with different suitability for meat production or flight. The breast muscle percentage in the carcasses of 28-week-old Cauchois × King and Wrocławski Meat × King squabs was 33.9% and 31.9%, respectively, which is higher than in the pigeons after three reproductive seasons from our study. As in the earlier studies, the leg muscle content was found to be low, which is related to the evolutionary adaptation of the pigeons [5]. In the study by Jiang et al. [4], leg muscles accounted for 6.83% to 7.97% of the weight of carcasses in 28-day-old White King pigeons, whereas in the study of Miąsko and Łukasiewicz [5], the values were 7.29% in 28-week-old Cauchois × King pigeons and 8.08% in Wrocławski Meat × King pigeons, respectively. The carcasses of the carrier and King pigeons evaluated in our study contained more abdominal fat and wings, and less neck compared to Egyptian pigeons aged 28 days [10].

The present study also determined the chemical composition of the breast and leg muscles. The content, balancing, and bioavailability of nutrients—protein, fat, minerals, and vitamins—determine the nutritive value of meat, which is considered one of the essential aspects of meat quality. According to Abdel-Azeem et al. [10], in some countries, medical value is attributed to pigeon meat due to its high protein and low cholesterol content.

The compared pigeon genotypes were characterized by the high protein content of the breast and leg muscles. Higher protein content was observed in breast muscles than in leg muscles. Dal Bosco et al. [8] established lower protein content (from 20.91% to 21.50%) in White King breeder pigeons than in the pigeons studied here. El-Aziz and Abdel-Raheem [28] also observed lower protein content (from 18.54% to 19.71%) in the breast muscles of parent local Egyptian Baladi pigeons compared to the carrier and King pigeon reported here. Pomianowski et al. [6] found significant differences between Wrocławski and King pigeons in the protein content of breast muscles, which is consistent with our study. Wrocławski pigeons showed a higher protein content in breast muscles (23.61%) compared to King pigeons (21.73%). In the same study, Europigeon breeds showed an intermediate protein content in breast muscle (23.16%), and the differences in relation to Wrocławski and King pigeons were not statistically significant. Another study [29] found higher protein content in the breast muscles of King (19.0%) and Strasser (18.5%) pigeons than in the muscles of Wrocławski × King pigeons (17.6%) and Wrocławski Meat × Strasser (18.1%) pigeons at 28 days of age. Apata et al. [30] reported that the protein content is higher in male than in female breast muscles, which agrees with our findings. The compared pigeon genotypes differed in the fat content of breast and leg muscles. The lower fat content in breast and leg muscles of the carrier pigeons in our study was likely related to their higher physical activity than King pigeons, which show poor flying ability and land mobility. The higher fat content of the meat from King pigeons compared to carrier pigeons may also result from changes in the metabolism of King pigeons in response to selection for rapid growth, or from the different expression of genes responsible for body fatness in the compared groups of birds [31].

El-Aziz and Abdel-Raheem [28] reported higher fat content in the breast muscle of parent local Egyptian Baladi pigeons (from 5.95% to 7.42%), whereas Pomianowski et al. [6] observed more fat in the breast muscle of Wrocławski pigeons (7.07%) aged 28 days. The experiment of Zieleziński et al. [29] noted higher fat content in the breast muscle of 28-day-old King (5.18%), Strasser (5.66%), Wrocławski Meat (6.27%), Strasser × King (5.60%), and Wrocławski Meat × King pigeons (5.67%) than in the carrier and King pigeons after three reproductive seasons. Bu et al. [17], who evaluated meat quality in White King pigeons at 28 and 600 days, found no significant changes in breast muscle water content with advancing age, as well as higher protein and fat content in the muscles of old pigeons compared to squab pigeons. Regardless of the genetic origin, the breast muscles of the studied pigeons contained more water and protein, and less fat, than leg muscles, which is in agreement with earlier studies [30]. Moreover, Zieleziński et al. [29] noted the distinctly lower collagen content (0.36 or 0.75%) in meat pigeons aged 28 days compared to the pigeons in our study.

An important indicator of meat quality is acidity, which has an effect on meat characteristics such as color, water holding capacity, taste, tenderness, cooking loss, and drip loss [32,33]. High meat acidity values (pH_24_) may indicate that the pigeons were skittish and their breast muscles were highly active before slaughter. Dong et al. [34] found lower pH_30_ values for the breast muscles (pH = 5.78–5.86) of 28-day-old pigeons, while Apata et al. [30] found higher pH values (males 6.86, females 6.80) in adult pigeons.

Another parameter that has never been determined in pigeons is the electrical conductivity of the meat. After slaughter, normal meat (without quality defects) is characterized by low electrical conductivity. This increases during storage. In our study, the breast muscles of King pigeons showed significantly higher electrical conductivity values measured 24 h postmortem compared to the EC_24_ values of the breast muscles from carrier pigeons. EC_24_ values for the breast muscle of the studied pigeons were lower than in six-week-old broiler chickens [35], and higher than in 112-week-old Pekin ducks [36].

Another important meat quality attribute is color. This trait has a key role during meat purchasing. It is considered to reflect the freshness and suitability of meat for certain culinary purposes. Our study found a significant effect of pigeon genotype on the lightness (L*) and yellowness (b*) of the pectoralis major muscle. The lighter color of breast muscle in King pigeons was probably associated with the higher fat content of breast muscles compared to the muscles of carrier pigeons. Higher motor activity of carrier pigeons breast muscle compared to King pigeons, which are characterized by poor flying ability, resulted probably in a higher percentage of dark, oxidative muscles, which determine higher a* (redness) values and lower color lightness expressed by the L* values. Jiang et al. [4] observed a darker color (lower L*, higher a*) in the breast muscles of 28-day-old White King pigeons compared to the pigeons from our study. Dong et al. [34] noted lower L* values (from 40.37 to 41.66) and higher b* values (from 14.00 to 17.40) in 25-day-old pigeons compared to the birds in our study. This was probably due to the lower susceptibility of older birds to preslaughter stress in our study, which contributed to better exsanguination and lower hemoglobin content, resulting in the lighter meat color of the pigeons compared to younger squab pigeons studied by Jiang et al. [4]. In turn, Bu et al. [17] found higher redness (a*) and yellowness (b*) values in White King pigeons aged 600 days compared to 28-day-old birds. Jijang et al. [4] observed higher chroma (C* = 20.84 to 24.09) and a generally lower hue angle (h* = 19.34⁰ to 21.45⁰) of breast muscles from 28-day-old White King pigeons than in our study. In another experiment [20], increased age led to more intense color (low h* value) and darker meat (high a*, b*, and C*) compared to younger animals. The significantly higher chroma (C*) and hue angle (h*) values of the breast muscles of carrier pigeons were associated with significantly lower yellowness (b*) and higher redness (a*) values compared to the muscles of King pigeons. The higher h* values of breast muscles from King pigeons compared to carrier pigeons indicate a less red and more discolored lean [20].

Our study was the first to determine the textural traits (hardness, springiness, cohesiveness, chewiness, and gumminess) and rheological properties (sum of elastic moduli and sum of viscous moduli) of the pectoralis major muscle in carrier and King pigeons. Differences in the mechanical properties of muscles (texture and relaxation test parameters) between the results obtained in this experiment and the data presented in the literature can be associated with both species differences, age of animals, and be related to the effect of the environmental conditions in which the animals lived (method of feeding, muscular workload). For example, Balowski et al. [37] found higher hardness (43.58 N) and lower springiness (1.0 cm), cohesiveness (0.248), chewiness (10.95 N × cm), and gumminess (10.9 N) of the pectoralis major muscle in hunted male wood pigeons (*Columba palumbus*) older than 12 months than in the farmed carrier pigeons (a variety of rock pigeon, *Columba livia*) and King pigeons studied here. In another experiment [34], shear force of breast muscles from 25-day-old Taishen King pigeons ranged from 14.33 to 18.22 N. In turn, Jiang et al. [4] observed in White King pigeons aged 28 days that shear force of breast muscles ranged from 21.61 to 23.11 N. Squabs are slaughtered at the age of 25–28 days when they have attained adult size, but have not yet flown. For these reasons, breast meat from squabs is very delicate and tender. In our study, we obtained distinctly higher shear force values for breast muscle of the pigeons after three reproductive seasons compared to young birds [4,34]. Probably the age of the pigeons and the method of shear force determination were the main factors affecting the high value of this trait. Bu et al. [17] reported higher shear force of breast muscles in 600-day-old pigeons than in squab pigeons aged 28 days. Male old pigeons had significantly higher shear force values of breast muscles compared to female squabs [17]. In the present paper, the value of WB test parameters was determined by cutting the meat parallel to the muscle fiber orientation, so it can be assumed that the connective tissue mainly determined the shear force value. This may result from the relatively high content of intramuscular connective tissue, which was determined in the present study based on the perimysium and endomysium thickness. Similar relationships were shown by Liu et al. [38], who concluded that chicken muscle hardness is significantly correlated to the total collagen content of the muscles and perimysium thickness.

The compared pigeon groups showed differences in the microstructural traits of the meat. Carrier pigeons had a significantly greater fiber cross-section area, fiber perimeter, and horizontal and vertical Feret diameter, which was related to the noticeably higher motor activity of their breast muscles compared to King pigeons, which are characterized by poor flying ability and land mobility. In male wood pigeons aged over 12 months, Balowski et al. [37] found a greater fiber cross-section area (389.4 μm^2^) and lower perimysium thickness (3.2 μm). The endomysium thickness of the breast muscle of wood pigeons was 1.01 μm, which is more than in King pigeons, but less than in the carrier pigeons studied here. However, the results obtained were significantly influenced by the age and origin of the birds.

In the present study, we also determined the dimensions of some internal organs of the carrier and King pigeons after three reproductive seasons. Abdel-Azeem et al. [10] found greater total intestinal length (from 127.4 to 129.0 cm) in 28-day-old squabs compared to the pigeons in our study. In turn, El-Eziz and Abdel-Raheem [27] reported lower weights of the gizzard (from 5.81 to 7.01 g) and heart (from 3.06 to 3.83 g) in parent local Egyptian Baladi pigeons aged 20‒22 months with an average body weight of 330 g compared to the carrier and King pigeons from our study. Lower liver weight (5.76 g) in five-year-old and older pigeons was observed by Kausar et al. [39] in Sialkoti pigeons. Miąsko and Łukasiewicz [5] reported differences in the weights of hearts, gizzards, and livers of 28-day-old Strasser pigeons and six hybrids created from the Cauchois, King, Polish Lynx, Wrocławski Meat, and Giant Homer breeds. Average liver, heart, and gizzard weights were highest in Giant Homer × King pigeons (11.5, 8.5, and 13.0 g, respectively), and lowest in Wrocławski Meat × King pigeons (8.0, 6.5, and 8.5 g, respectively).

## 5. Conclusions

In summary, after three reproductive seasons, the carrier and King pigeons differed significantly (*p* < 0.05) in terms of carcass weight and measurements, and in the percentage of leg muscles, skin with subcutaneous fat, abdominal fat, neck, wings, and remainders of eviscerated carcass. Pigeon genotype had a significant (*p* < 0.05) effect on basic chemical analysis, except collagen content in leg muscles, and also on electrical conductivity, lightness, yellowness, chroma and hue angle, hardness, springiness, chewiness, gumminess, sum of elastic and viscous moduli, and all the microstructural traits of the pectoralis major muscle. King pigeons were characterized by a significantly (*p* < 0.05) greater total length of the intestine and its segments, except for duodenal length, as well as higher weight of the proventriculus, gizzard, heart, liver, and spleen, and had a significantly (*p* < 0.05) smaller duodenal length and diameter compared to the carrier pigeons. The present study provides information about the carcass composition, nutritive, and physicochemical properties, and texture and microstructure of pigeon meat after three reproductive seasons, which could be useful for consumers of pigeon meat.

## Figures and Tables

**Table 1 animals-10-01315-t001:** Carcass weight and measurements in three-year-old pigeons of different genotypes.

Trait	King		Carrier Pigeon		SEM	*p*-Value		
	Male	Female	Male	Female		G	S	G × S
CW (g)	471.3	463.6	327.0	274.8	15.7	0.001	0.049	0.224
CL (cm)	20.8	20.7	18.2	18.0	0.3	0.001	0.600	0.916
TL (cm)	12.4	12.4	12.8	12.3	0.1	0.001	0.603	0.917
CC (cm)	22.5	22.4	20.7	19.8	0.2	0.001	0.049	0.389
KL (cm)	10.3	9.8	9.0	8.8	0.1	0.001	0.578	0.131
DL (cm)	8.7	8.4	7.4	7.1	0.1	0.001	0.184	0.184

G—Genotype, S—Sex, G × S—Genotype by sex interaction; *n* = 40 for each analyzed trait, CW—Carcass weight, CL—Carcass length, TL—Trunk length, CC—Chest circumference, KL—Keel length, DL—Drumstick length.

**Table 2 animals-10-01315-t002:** Share (%) of carcass components in three-year-old pigeons of different genotypes.

Trait	King		Carrier Pigeon		SEM	*p*-Value		
	Male	Female	Male	Female		G	S	G × S
BM (%)	30.6	28.7	29.9	30.5	0.5	0.501	0.185	0.418
LM (%)	6.9	7.1	6.5	5.5	0.2	0.007	0.107	0.435
SF (%)	10.0	12.9	15.2	15.2	0.8	0.002	0.806	0.466
AF (%)	1.9	1.9	1.6	1.6	0.2	0.004	0.676	0.515
NC (%)	3.8	3.2	2.8	2.7	0.1	0.001	0.019	0.069
WI (%)	17.3	18.1	15.9	15.7	0.3	0.001	0.943	0.397
CR (%)	29.5	28.1	28.1	28.8	0.1	0.011	0.767	0.756

G—Genotype, S—Sex, G × S—Genotype by sex interaction; *n* = 40 for each analyzed trait, BM—Breast meat, LM—Leg meat, SF—Skin with subcutaneous fat, AF—Abdominal fat, NC—Neck, WI—Wings, CR—Carcass remainders.

**Table 3 animals-10-01315-t003:** Chemical composition of the breast and leg meat in three-year-old pigeons of different genotypes.

Trait		King		Carrier Pigeon		SEM	*p*-Value		
		Male	Female	Male	Female		G	S	G × S
Water (%)	BM	66.5	66.5	68.1	69.2	0.2	0.001	0.185	0.133
	LM	62.1	61.5	65.3	65.1	0.3	0.001	0.245	0.623
Protein (%)	BM	27.5	26.3	24.7	24.3	0.2	0.010	0.608	0.325
	LM	21.2	20.5	22.5	22.9	0.2	0.001	0.498	0.049
Fat (%)	BM	4.2	5.0	2.9	2.0	0.2	0.010	0.880	0.023
	LM	11.9	13.3	8.0	7.6	0.4	0.001	0.395	0.122
Collagen (%)	BM	1.9	2.9	1.6	2.0	0.1	0.001	0.001	0.027
	LM	2.1	1.9	2.1	2.1	0.1	0.144	0.169	0.020

G—Genotype, S—Sex, G × S—Genotype by sex interaction; *n* = 40 for each analyzed trait, BM—Breast muscles, LM—Leg muscles.

**Table 4 animals-10-01315-t004:** Selected physicochemical traits of the pectoralis major muscle in three-year-old pigeons of different genotypes.

Trait	King		Carrier Pigeon		SEM	*p*-Value		
	Male	Female	Male	Female		G	S	G × S
pH_24_	6.33	6.30	6.36	6.41	0.1	0.060	0.409	0.122
EC_24_ (mS/cm)	8.06	8.96	6.89	8.12	0.2	0.028	0.082	0.711
L*—lightness	47.1	47.8	45.7	42.8	0.8	0.018	0.110	0.072
a*—redness	15.0	14.6	14.4	16.0	0.3	0.592	0.385	0.159
b*—yellowness	9.6	9.7	7.0	5.8	0.4	0.001	0.166	0.161
C*—chroma	17.8	17.5	16.0	17.0	0.4	0.015	0.754	0.147
h*—hue angle (⁰)	32.6	33.6	25.9	19.9	0.9	0.001	0.004	0.450

G—Genotype, S—Sex, G × S—Genotype by sex interaction; *n* = 40 for each analyzed trait.

**Table 5 animals-10-01315-t005:** Texture and rheological properties of the pectoralis major muscle in three-year-old pigeons of different genotypes.

Trait	King		Carrier Pigeon		SEM	*p*-Value		
	Male	Female	Male	Female		G	S	G × S
Hardness (N)	35.3	39.5	34.9	30.1	1.0	0.004	0.655	0.021
Cohesiveness	0.4	0.4	0.4	0.4	0.1	0.127	0.965	0.536
Springiness (cm)	1.4	1.5	1.3	1.4	0.1	0.009	0.156	0.222
Chewiness (N × cm)	21.3	23.6	17.7	15.4	0.9	0.001	0.882	0.183
Gumminess (N)	15.2	16.9	13.6	11.8	0.6	0.001	0.725	0.180
WB shear force (N)	95.1	96.1	112.3	91.8	8.4	0.835	0.300	0.822
EMS (kPa)	380.0	486.0	377.0	323.0	13.0	0.001	0.359	0.001
VMS (kPa × s)	16472	23325	17019	13440	866.9	0.001	0.352	0.001

G—Genotype, S—Sex, G × S—Genotype by sex interaction; *n* = 40 for each analyzed trait, EMS—Sum of elastic moduli, VMS—Sum of viscous moduli.

**Table 6 animals-10-01315-t006:** Microstructure of the pectoralis major muscle in three-year-old pigeons of different genotypes.

Trait	King		Carrier Pigeon		SEM	*p*-Value		
	Male	Female	Male	Female		G	S	G × S
Fiber cross-section area (μm^2^)	108.1	96.2	121.8	128.0	4.4	0.006	0.866	0.361
Fiber perimeter (μm)	42.8	41.1	47.9	48.7	1.0	0.002	0.701	0.593
Fiber diameter H (μm)	11.3	10.8	12.6	12.9	0.3	0.001	0.780	0.522
Fiber diameter V (μm)	11.7	11.0	12.9	13.1	0.3	0.003	0.838	0.484
H:V diameter ratio (x)	0.97	0.98	0.98	0.98	0.1	0.881	0.892	0.791
Perimysium thickness (μm)	5.0	5.0	5.4	6.0	0.2	0.028	0.334	0.494
Endomysium thickness (μm)	0.9	0.9	1.2	1.3	0.1	0.035	0.717	0.831

G—Genotype, S—Sex, G × S—Genotype by sex interaction; *n* = 40 for each analyzed trait.

**Table 7 animals-10-01315-t007:** Biometric traits of some organs in three-year-old pigeons of different genotypes.

Trait	King		Carrier Pigeon		SEM	*p*-Value		
	Male	Female	Male	Female		G	S	G × S
Length (cm)								
Total intestine	118.0	122.9	105.3	105.5	2.3	0.001	0.908	0.712
Duodenum	13.3	13.8	17.8	15.2	0.4	0.001	0.102	0.021
Jejunum + Ileum	99.7	103.2	83.5	86.7	2.2	0.001	0.877	0.821
Caeca	0.6	0.4	0.4	0.4	0.1	0.007	0.001	0.002
Colon	4.4	5.5	3.6	3.2	0.2	0.001	0.393	0.052
Diameter (mm)								
Duodenum	6.5	6.5	8.8	7.6	0.3	0.001	0.109	0.085
Jejunum + Ileum	4.3	4.8	4.9	4.2	0.1	0.674	0.595	0.049
Colon	3.8	3.6	3.6	3.7	0.1	0.694	0.890	0.413
Weight (g)								
Proventriculus	1.3	1.5	0.9	0.8	0.1	0.001	0.661	0.380
Gizzard	11.8	12.3	8.2	7.1	0.5	0.001	0.106	0.431
Liver	13.8	12.7	6.6	6.1	0.2	0.001	0.252	0.456
Heart	7.3	6.3	5.6	5.7	0.7	0.001	0.143	0.168
Spleen	0.4	0.4	0.2	0.3	0.1	0.002	0.412	0.038

G—Genotype, S—Sex, G × S—Genotype by sex interaction; *n* = 40 for each analyzed trait.

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
