# Peer review of "Carcass Characteristics, Physicochemical Properties, and Texture and Microstructure of the Meat and Internal Organs of Carrier and King Pigeons"

_animals, 2020, doi:10.3390/ani10081315_

Round 1
Reviewer 1 Report
I read with great interest the paper by Kokoszyński and co-workers. The papers had some drawbacks that need to be fixed before extensive review.
I recommend to the authors to carefully revise their paper and remove the redundancy. For example, the protocols of meat quality evaluation, especially texture, are not very clear and repetition/confusion is easily detectable. Please clarify as much as possible the details of each protocol. Separate the analyses by sub-titles. For colour, I recommend to the authors to compute hue angle (h*) and Chroma (C*), include them in the analyses and discussion. Refer to this paper for help to compute them: https://doi.org/10.1016/j.meatsci.2018.03.004
Also, the statistics are not complete. I suggest multivariate analyses, such as Principal Component Analysis by spotting all the groups that you have with all variables. It is not clear if the n=40 are always the same or no. Please clarify this in the manuscript including the figure captions.
I have a concern about the shear force values that are very high. Accordingly, I recommend to the authors to shown the descriptive statistics of all the variables investigate din this paper. Put this as supplementary. Raw data are welcome to check the accuracy of the results.
The English of the papers needs improvement.
Genotype (G) – sex (S) is not very clear. Please clarify what is the difference.
Author Response
Answers to the comments of Reviewer no. 1
Comment 1
Extensive editing of English language and style required
Answer
After the revision, the paper was sent to MDPI English editing service for English language correction.
Comment 2
I recommend to the authors to carefully revise their paper and remove the redundancy. For example, the protocols of meat quality evaluation, especially texture, are not very clear and repetition/confusion is easily detectable. Please clarify as much as possible the details of each protocol. Separate the analyses by sub-titles
Answer
L182-223
A new detailed text concerning the determination of textural characteristics and rheological properties was introduced.
2.4. Meat texture
The textural traits (hardness, cohesiveness, chewiness, springiness gumminess, Warner‒Bratzler shear force) of the 40 samples of processed pectoralis major muscle of pigeons after three reproductive seasons were determined using an Instron 1140 machine (Instron Corp., Norwood, MA, USA). The tests were performed on heat-treated samples. The samples of meat were tightly packed into plastic bags and heated in water at 72 ± 2 °C until the internal temperature reached 70.2 °C in the geometric centre of the sample. Next, the samples were chilled in water to 20 °C, wrapped in food-grade plastic film, stored for 12 h in a refrigerator at 4 °C, and heated to 18 °C. After removing the wrap from each sample of the pectoralis major muscle, slices of 20 ± 1 mm thickness were cut perpendicular to the muscle fibre orientation using a Siemens MS600 electric slicer (Hausgeräte GmbH, Köln, Germany). The prepared meat samples were measured for textural traits using the Texture Profile Analysis (TPA) and WB (Warner‒Bratzler) tests.
The TPA test involved twice pushing a plunger of 0.96 cm diameter into a sample 20 ± 1 mm high, to 80% (16 mm) of its depth. Using the force‒deformation curve relationship, the following texture parameters were calculated: hardness—maximum height of the first peak, cohesiveness—ratio of the second peak area to the first peak area, springiness—width of the base of the ascending portion of the second peak, gumminess—product of hardness and cohesiveness, and chewiness—a product of hardness, cohesiveness, and springiness [20]. For each sample, the TPA test was repeated four times and the total number of determinations was 160.
The Warner‒Bratzler (WB) test consisted of cutting, transversely across the muscle fibres, a sample with a 10 x 10 mm cross-sectional area using a triangular blade. A crosshead speed of 50 mm/min was applied. This test was used to determine the trait WB shear force.
2.5. Rheological properties
The rheological properties (sum of viscous moduli and sum of elastic moduli) of the 40 samples of processed pectoralis major muscle of pigeons were determined using an Instron 1140 machine (Instron Corp.). During the relaxation test, a plunger of 1.26 cm diameter was pushed into the muscle samples 2 mm deep while recording for 90 s the changes in tension. The springiness and viscosity moduli were calculated using the generalized Maxwell model, consisting of three elements connected in parallel: Hooke’s body and Maxwell’s two viscoelastic bodies. The model’s equation is:
,
where:
δ—tension (kPa)
ε—deformation
E0—elastic modulus of Hooke’s body (kPa)
E1,E2—elastic moduli of Hooke’s body 1 and 2 (kPa)
µ1, µ2—viscous moduli of Maxwell’s body 1 and 2 (kPa x s)
t—time
To make the interpretation of the results easier, the sum of elastic moduli (E0 + E1 + E2) and the sum of viscous moduli (µ1 + µ2) were calculated for each sample.
Comment 3
Separate the analyses by sub-titles
Answer
The following subtitles were added to the “Materials and Methods”:
2.1. Carcass collection and evaluation of carcass traits
2.2. Basic chemical composition
2.3. Physicochemical characteristics
2.4. Meat texture
2.5. Rheological properties
2.6. Microstructure of meat
2.7. Digestive system characteristics
2.8. Statistical analysis
Comment 4
For colour, I recommend to the authors to compute hue angle (h*) and Chroma (C*), include them in the analyses and discussion. Refer to this paper for help to compute them: https://doi.org/10.1016/j.meatsci.2018.03.004
Answer
The following new text was added to the manuscript:
L176-178 (in the Materials and Methods section)
The redness (a*) and yellowness (b*) values were used to calculate chroma (C*) and hue angle (h*). Chroma (C*) was calculated with the formula C = (a*2 + b*2)1/2, and hue angle (h*) with the formula arctg (b*/a*).
L316 (In Results section)
“chroma and hue” was added
L43 (in Abstract section)
“chroma (C*) and hue angle (h*)” was added after “yellowness (b*)”
L509 (in Conclusions section)
“chroma and hue angle” was added after “yellowness”
L461-464 (in Discussion section)
Jijang et al. [4] observed higher chroma (C* = 20.84 to 24.09) and a generally lower hue angle (h* = 19.34⁰ to 21.45⁰) than in our study. The significantly higher chroma (C*) and hue angle (h*) values of the breast muscles of carrier pigeons were associated with significantly lower yellowness (b*) and higher redness (a*) values compared to the muscles of King pigeons.
In Table 4
The numerical data for C* and h* were added
King Carrier pigeon SEM P-value
Male Female Male Female G S G x S
C* - chroma 17.8 17.5 16.0 17.0 0.4 0.015 0.754 0.147
h- hue angle (⁰) 32.6 33.6 25.9 19.9 0.9 0.001 0.004 0.450
Comment 5
Also, the statistics are not complete. I suggest multivariate analyses, such as Principal Component Analysis by spotting all the groups that you have with all variables.
Answer
Thank you for drawing attention to the possibility of data analysis using the PCA method. We think, however, that in the case of our experiment the optimal solution is two-ay analysis of variance with the main factors breed and sex, and the interaction between them. We would like to point out that the applied statistical approach is commonly used for the analysis of poultry performance data in the studies by other authors and in our studies, e.g.:
- Uhlirová L, Tůmová E, Chodová D, Vičkova J, Ketta M, Volek Z, Skŕivanova V. 2018. The effect of age, genotype and sex on carcass traits, meat quality and sensory attributes of geese. Asian-Australas J Anim Sci 31:421-428.
- Yamak US, Sarica M, Boz MA, Ucar A. 2020. Effect of production system and age on the growth performance and carcass traits of Pheasants (Phasianus colchicus), Ann. Anim. Sci., 20, 1, 219-229.
- Kokoszyński D., Wasilewski R., Steczny K., Kotowicz M., Hrnčar C, Arpášova H. (2019). Carcass composition and selected meat quality traits of Pekin ducks from genetic resources flocks. Poult. Sci. 98(7): 3029-3039.
Comment 6
It is not clear if the n=40 are always the same or no. Please clarify this in the manuscript including the figure captions.
Answer
The following explanation was inserted under Tables 1-7:
n=40 for each analysed trait
And a sentence was added in the Materials and Methods
L148-149: “40” was added before breast/leg/ pectoral major.
Comment 7
I have a concern about the shear force values that are very high. Accordingly, I recommend to the authors to shown the descriptive statistics of all the variables investigate din this paper. Put this as supplementary. Raw data are welcome to check the accuracy of the results.
Answer:
The raw data for WB-shear force were reviewed. The values for males were corrected. The WB-shear force values for female no. 614 (carrier pigeons) were disregarded due to the high shear force value (which was markedly higher than others) resulting probably from the large amount of connective tissue.
The data for meat texture determination are detailed in the Supplementary material attached to the manuscript.
The following text was added to the Discussion:
L471-481
Texture depends on structural elements and their organization [38]. Papa and Fletcher [39] report that the thickness and number of muscle fibres per bundle influence meat texture, and, according to Palka [40], Nishimura et al. [41], and Purslow [42], the composition, amount, and distribution of connective tissue determine meat hardness and the force required to shear the meat. In the present paper, the value of WB test parameters was determined by cutting the meat parallel to the muscle fibre orientation, so it can be assumed that the connective tissue mainly determined the shear force value. This may result from the relatively high content of intramuscular connective tissue, which was determined in the present study based on the perimysium and endomysium thickness. Similar relationships were shown by Liu et al. [43], who concluded that chicken muscle hardness is significantly correlated to the total collagen content of the muscles and perimysium thickness.
The following items were added to the References (nos. 37-42):
Brooks, J.C.; Savell J.W. Perimysium thickness as an indicator of beef tenderness. Meat Sci. 2004, 67, 329-334.
Papa, C.M.; Fletcher, D.L. Pectoralis muscle shortening and rigor development at different locations within broiler breast. Food Technol. 1988, 18, 848-902.
Palka, K. Structural basis for meat texture. Food Technology Quality 1995, 1, 8-16.
Nishimura, T.; Liu, A.; Hattori, A.; Takahashi, K. Changes in mechanical strength of intramuscular connective tissue during postmortem ageing of beef. J. Anim. Sci. 1988, 76, 528-532.
Purslow, P.P. Intramuscular connective tissue and its role in meat quality. Meat Sci. 2005, 70, 435-447.
Liu, A.; Nishimura, T.; Takahashi, K. Structural weakening of intramuscular tissue during post mortem aging of chicken semitendinosus muscle. Meat Sci. 1995, 39, 135-142.
Comment 8
The English of the papers needs improvement
Answer:
After the revision, the paper was sent to MDPI English editing service for English language correction.
Comment 9
Genotype (G) – sex (S) is not very clear. Please clarify what is the difference
Answer:
The text “Genotype (G) – sex (S)” was removed from the headings of Tables 1 to 7
The following description was added under Tables 1 to 7:
G - genotype, S – sex, G x S – genotype by sex interaction
The authors thank the Reviewer for in-depth review of the paper.

Reviewer 2 Report
differences in p value signature - line 273-275 p<0.05, in lines 296-303 P<0.05 (italics and capital letter)
line 220 - lack of dot (0.01 cm)
suggestion:
lines 427 - compared genotypes - unfortunate term - can be received very genetically
Author Response
Answers to the comments and suggestions of Reviewer no. 2
Comment 1
differences in p value signature - line 273-275 p<0.05, in lines 296-303 P<0.05 (italics and capital letter)
Answer
In lines 277-279, (p < 0.05) replaced (P < 0.05) , small “p” in italics
Comment 2
line 220 - lack of dot (0.01 cm)
Answer
Line 246 – dot was added
Suggestion 1
lines 427 - compared genotypes - unfortunate term - can be received very genetically
Line 482
“compared genotypes” was changed to “compared groups”
The authors thank the Reviewer for reviewing the paper.

Reviewer 3 Report
Overall, I have found the study well conducted and well written. It assess differences in pigeon selected for two different use.
Abstract:
Line 36-37: The sentence is confounding, please clarify.
Line 37: In my opinion a short sentence on what method was used to compare the two breed is needed.
Introduction: The introduction is overall informative and well-written. I would only suggest to add more description of carrier pigeon, since it is one of the main subject of your study.
Discussion:
I would suggest to state clearer which are the results of the present study and better integrate them in the comparison with the literature.
I would suggest also to add biological interpretation according to the existing literature, on why the two different genotypes present such differences. If not available literature exist, you may compare it with what has been known in closer species intended for meat production.
Conclusion:
I would like to see also a small recap on why the results of the present study are insight for the sector.
Author Response
Answers to the comments of Reviewer no. 3
Comment 1
English language and style are fine/minor spell check required
Answer
After the revision, the paper was sent to MDPI English editing service for English language correction.
Comment 2
Line 36-37: The sentence is confounding, please clarify.
Answer
L36-37
The following was removed: “including 10 male carcasses and 10 female carcasses each”
Comment 3
Line 37: In my opinion a short sentence on what method was used to compare the two breed is needed.
Answer
A new sentence was added in the Abstract.
L37-40
The basic chemical composition was determined by near-infrared transmission (NIT) spectroscopy, colour coordinates according to CIELab, the texture according to Texture Profile Analysis (TPA) and Warner–Bratzler (WB) tests, and the rheological properties of meat according to the relaxation test.
Comment 4
Introduction: The introduction is overall informative and well-written. I would only suggest to add more description of carrier pigeon, since it is one of the main subject of your study.
Answer
The following sentence was added in the “Introduction”;
L82-83
Carrier pigeons were once used to send messages. Today, their homing ability gives great satisfaction to breeders and is increasingly used during carrier pigeon flying competitions [3].
Comments 5 & 6
Discussion:
I would suggest to state clearer which are the results of the present study and better integrate them in the comparison with the literature.
I would suggest also to add biological interpretation according to the existing literature, on why the two different genotypes present such differences. If not available literature exist, you may compare it with what has been known in closer species intended for meat production.
Answer
The following text was added to the Discussion:
L456-459
This was probably due to the lower susceptibility of older birds to preslaughter stress in our study, which contributed to better exsanguination and lower haemoglobin content, resulting in the lighter meat colour of the pigeons compared to younger squab pigeons studied by Jiang et al. [4].
L418-421
The higher fat content of the meat from King pigeons compared to carrier pigeons may also result from changes in the metabolism of King pigeons in response to selection for rapid growth, or from the different expression of genes responsible for body fatness in the compared groups of birds [30].
The following item was added to the References:
Buzała, M.; Janicki, B. Review: Effects of different growth rates in broiler breeder and layer hens on some productive traits. Poult. Sci. 2016, 95, 2151-2159.
L 471-481
Texture depends on structural elements and their organization [38]. Papa and Fletcher [39] report that the thickness and number of muscle fibres per bundle influence meat texture, and, according to Palka [40], Nishimura et al. [41], and Purslow [42], the composition, amount, and distribution of connective tissue determine meat hardness and the force required to shear the meat. In the present paper, the value of WB test parameters was determined by cutting the meat parallel to the muscle fibre orientation, so it can be assumed that the connective tissue mainly determined the shear force value. This may result from the relatively high content of intramuscular connective tissue, which was determined in the present study based on the perimysium and endomysium thickness. Similar relationships were shown by Liu et al. [43], who concluded that chicken muscle hardness is significantly correlated to the total collagen content of the muscles and perimysium thickness.
The following items were added to the References (nos. 37-42):
Brooks, J.C.; Savell J.W. Perimysium thickness as an indicator of beef tenderness. Meat Sci. 2004, 67, 329-334.
Papa, C.M.; Fletcher, D.L. Pectoralis muscle shortening and rigor development at different locations within broiler breast. Food Technol. 1988, 18, 848-902.
Palka, K. Strukturalne podstawy tekstury mięsa. Food Technology Quality 1995, 1, 8-16.
Nishimura, T.; Liu, A.; Hattori, A.; Takahashi, K. Changes in mechanical strength of intramuscular connective tissue during postmortem ageing of beef. J. Anim. Sci. 1988, 76, 528-532.
Purslow, P.P. Intramuscular connective tissue and its role in meat quality. Meat Sci. 2005, 70, 435-447.
Liu, A.; Nishimura, T.; Takahashi, K. Structural weakening of intramuscular tissue during post mortem aging of chicken semitendinosus muscle. Meat Sci. 1995, 39, 135-142.
Comment 7
Conclusion:
I would like to see also a small recap on why the results of the present study are insight for the sector
Answer
L 514-516
The present study provides information about the carcass composition, nutritive, and tphysicochemical properties, and texture and microstructure of pigeon meat after three reproductive seasons, which could be useful for consumers of pigeon meat.
The authors thank the Reviewer for reviewing the paper.

Round 2
Reviewer 1 Report
The authors improved the quality of their paper.
I asked in my previous review to cite a reference for color measurmemnts (https://doi.org/10.1016/j.meatsci.2018.03.004) and also for all the other protocols used.
Further, I pointed in my past review the high values of shear forces and the authors did not explain this in the manuscript.
The discussion still poor and needed realed discussion and not comparisons to papers or reviews that are not dealing with pigeons meat. This has to be considered. The new references added in red are not appropariate such as that one by Purslow.
Author Response
Answers to the comments and suggestions of Reviewer no. 1
Comment 1
English language and style are fine/minor spell check required
Answer
The paper was sent to MDPI English editing service for English language correction.
Comment 2
I asked in my previous review to cite a reference for color measurmemnts (https://doi.org/10.1016/j.meatsci.2018.03.004) and also for all the other protocols used.
Answer
L178: [20] was added
L584-585, the following was added:
- Gagaoua, M.; Picard, B.; Monteils, V. Associations among animal, carcass, muscle characteristics, and fresh meat color traits in Charolais cattle. Meat Sci. 2018, 140, 145-156.
L468-469, the following text was added:
In another experiment [20], increased age led to more intense colour (low h* value) and darker meat (high a*, b*, and C*) compared to younger animals.
L472-473
The higher h* values of breast muscles from King pigeons compared to carrier pigeons indicate a less red and more discoloured lean [20].
Comment 3
Further, I pointed in my past review the high values of shear forces and the authors did not explain this in the manuscript.
Answer
L484-494
In another experiment [34], shear force of breast muscles from 25-day-old Taishen King pigeons ranged from 14.33 to 18.22 N. In turn, Jiang et al [4] observed in White King pigeons aged 28 days that shear force of breast muscles ranged from 21.61 to 23.11 N. Squabs are slaughtered at the age of 25-28 days when they have attained adult size, but have not yet flown. For these reasons, breast meat from squabs is very delicate and tender. In our study, we obtained distinctly higher shear force values for breast muscle of the pigeons after three reproductive seasons compared to young birds [4,34]. Probably the age of the pigeons and the method of shear force determination were the main factors affecting the high value of this trait. Bu et al. [17] reported higher shear force of breast muscles in 600-day-old pigeons than in squab pigeons aged 28 days. Male old pigeons had significantly higher shear force values of breast muscles compared to female squabs [17].
In addition, in the previous answer to the review I included a description concerning the effect of perimysium and endomysium thickness of the pectoralis major muscle from the studied groups of pigeons on shear force value (L494-500).
Comment 4
The following text was removed as it was inappropriate:
“Texture depends on structural elements and their organization [38]. Papa and Fletcher [39] report that the thickness and number of muscle fibres per bundle influence meat texture, and, according to Palka [40], Nishimura et al. [41], and Purslow [42], the composition, amount, and distribution of connective tissue determine meat hardness and the force required to shear the meat”
The following new text was added in the Discussion”
L455-458
Higher motor activity of carrier pigeons breast muscle compared to King pigeons, which are characterized by poor flying ability, resulted probably in a higher percentage of dark, oxidative muscles, which determine higher a* (redness) values and lower colour lightness expressed by the L* values.
L468-469
In another experiment [20], increased age led to more intense colour (low h* value) and darker meat (high a*, b*, and C*) compared to younger animals.
L472-473
The higher h* values of breast muscles from King pigeons compared to carrier pigeons indicate a less red and more discoloured lean [20].
L476-480
Differences in the mechanical properties of muscles (texture and relaxation test parameters) between the results obtained in this experiment and the data presented in the literature can be associated with both species differences, age of animals, and be related to the effect of the environmental conditions in which the animals lived (method of feeding, muscular work load).
L484-494
In another experiment [34], shear force of breast muscles from 25-day-old Taishen King pigeons ranged from 14.33 to 18.22 N. In turn, Jiang et al [4] observed in White King pigeons aged 28 days that shear force of breast muscles ranged from 21.61 to 23.11 N. Squabs are slaughtered at the age of 25-28 days when they have attained adult size, but have not yet flown. For these reasons, breast meat from squabs is very delicate and tender. In our study, we obtained distinctly higher shear force values for breast muscle of the pigeons after three reproductive seasons compared to young birds [4,34]. Probably the age of the pigeons and the method of shear force determination were the main factors affecting the high value of this trait. Bu et al. [17] reported higher shear force of breast muscles in 600-day-old pigeons than in squab pigeons aged 28 days. Male old pigeons had significantly higher shear force values of breast muscles compared to female squabs [17].
The References number were corrected throughout the text.
The authors thank the Reviewer for the insightful review.
